# Coordination Chemistry of Polynitriles, Part XII—Serendipitous Synthesis of the Octacyanofulvalenediide Dianion and Study of Its Coordination Chemistry with K⁺ and Ag⁺

Patrick Nimax, Yannick Kunzelmann and Karlheinz Sünkel *

Department Chemistry, Ludwig-Maximilians University Munich, Butenandtstr. 5-13, 81377 Munich, Germany
* Correspondence: suenk@cup.uni-muenchen.de

**Abstract:** The reaction of diazotetracyanocyclopentadiene with copper powder in the presence of $NEt_4Cl$ yields, unexpectedly, besides the known $NEt_4[C_5H(CN)_4]$ (**3**), the $NEt_4$ salt of octacyanofulvalenediide $(NEt_4)_2[C_{10}(CN)_8]$ (**5**), which can be transformed via reaction with $AgNO_3$ to the corresponding $Ag^+$ salt (**4**), which in turn can be reacted with KCl to yield the corresponding $K^+$ salt **6**. The molecular and crystal structures of **4–6** could be determined, and show a significantly twisted aromatic dianion which uses all its nitrile groups for coordination to the metals; **4** and **6** form three-dimensional coordination polymers with fourfold coordinated $Ag^+$ and eightfold coordinated $K^+$ cations.

**Keywords:** diazotetracyanocyclopentadiene; octacyanofulvalenediide; radical chemistry; coordination polymers

## 1. Introduction

Diazotetracyanocyclopentadiene (**1**) was first reported by Webster in 1965. Reactivity studies showed that it "is a diazonium rather than a diazo compound", and that the mechanism of its reactions "is likely free radical". The presumed tetracyanocyclopentadienyl radical intermediate could be "generated polarographically at 0.23 V vs. sce in acetonitrile or by mild reducing agents" [1]. Radical reactions are in general typical of arenediazonium ions and allow "an easy entry into the chemistry of the aryl radical". For the generation of the radical, a reducing agent is needed, and Cu(I) appeared to be the most popular one. Typical reactions of the produced aryl radicals include H atom abstractions, formal halogen atom additions (so-called "Sandmeyer reactions") and homo- and hetero-aryl coupling reactions [2]. Although this type of chemistry is very old, it is still being studied intensively, and new synthetic applications are being developed [3,4]. In the chemistry of the diazotetracyanocyclopentadiene molecule, one important difference compared to the just mentioned arenediazonium ions is that this compound is neutral, and therefore one-electron reduction generates a radical anion instead of a neutral radical. A combined theoretical/electrochemical study showed that this radical anion can be protonated to give the $[C_5H(CN)_4]$ radical, which in turn can be reduced to the $[C_5H(CN)_4]$ anion, or vice versa [5]. Calculations showed that the neutral radicals $[C_5X(CN)_4]$ (X = H, CN) are strong oxidizers and can be termed "superhalogens" [6] or "hyperhalogens" [7], which is by definition an atom or atom group that has an electron affinity higher than a halogen atom. We could show recently that the radicals $[C_5X(CN)_4]$ can be isolated and structurally characterized, when X = $NH_2$ [8], while for X = $NO_2$ only electrochemical and EPR characterization was possible [9]. Treatment of the $NH_2$ or $NO_2$- substituted anions with Fe(III) generated Fe(II), and another group showed that when trying to prepare a Cu(II) complex of the $[C_5(CN)_5]$ anion, a "yet unknown oxidation of the $[C_5(CN)_5]$ ligand"

occurred [10]. The fate of the supposed intermediate $[C_5X(CN)_4]$ radicals remained unclear. For halogen radicals, dimerization to obtain the dihalogen $X_2$ molecules is common text-book knowledge. Thus, a dimerization of "superhalogens" might be expected as well. For radical anions such as $[TCNQ]^-$, a σ-dimerization to give the $[TCNQ\text{-}TCNQ]^{2-}$ dianion has been reported [11], while for $[DDQ]^{-\cdot}$ a π dimer with formation of "pancake bonds" was observed [12,13].

Although there are many hints to the formation of radicals in the chemistry of the diazotetracyanocyclopentadiene molecule, one should not forget that many diazocyclopentadienes show the typical carbene chemistry. Thus, low-temperature photolysis of $[C_5Br_4N_2]$ in an argon matrix generates the tetrabromocyclopentadienylidene carbene, which reacts with CO to give a ketene. Dimerization to octabromofulvalene is not observed at this temperature [14]. Room-temperature laser flash photolysis of $[C_5Cl_4N_2]$ in several solvents, such as alcohols, pyridine or THF, produced ylidic products derived from a triplet carbene [15], while low-temperature irradiation in an argon matrix in the presence of $CF_3I$ produced an ylidic iodonium ion [16]. Thermolysis of $[C_5Cl_4N_2]$ in the absence of solvent produced octachloronaphthalene, while treatment of hexane solutions of $[C_5X_4N_2]$ (X = Br, Cl) with bis(μ-chloro-π-allyl palladium) at 0 °C generated the octahalofulvalenes. Both product types were regarded as products of carbene intermediates [17]. Reaction of these tetrahalo-diazocyclopentadienes with some Ni(0), Pt(0) and Ru(0) coordination compounds produced complexes of the type "$L_nM(N_2\text{-}C_5X_4)$", where the diazo compound is coordinated by one or both diazo-nitrogens [18]. A high-temperature–high-pressure study of compound **1** in hexafluoro-isopropanol solution, devoted to the study of possible dinitrogen exchange, lead to the conclusion that the dediazoniation of (**1**) leads to an $^1A_2$ singlet carbene [19]. Formation of a dimerization product such as octacyanofulvalene or octacyanonaphthalene was not reported. However, thermal decomposition of $Ag[C_5H(CN)_2(OMe)_2]$ produced the corresponding fulvalene $[C_{10}(CN)_4(OMe)_4]$, which was also shown to undergo a stepwise two-electron reduction to the corresponding fulvalene dianion [20,21].

During the course of our studies on the coordination chemistry of $[C_5X(CN)_4]$ ions, we wanted to also look at the bromo and chloro derivatives (X = Br, Cl). For this purpose, we had to prepare these anions first, and we decided to employ the original synthetic pathway described by Webster in 1966. Here, we describe our experiences with this literature procedure and the unexpected results we obtained.

## 2. Results

### 2.1. Synthesis

2.1.1. Reaction of Diazotetracyanocyclopentadiene with Chloride and Bromide

The thermal reaction of diazotetracyanocyclopentadiene (1) with $NEt_4^+\,Cl^-$ in the presence of copper powder or with $NEt_4^+\,Br^-$ without additives was reported by Webster to yield the halo-tetracyanocyclopentadienides 2a/b in high yields ([1], Scheme 1).

**Scheme 1.** Synthesis of the halo-tetracyanocyclopentadienides **2a/b** according to Webster.

When we tried to repeat the synthesis of **2a**, we obtained a product mixture. The $^1$H NMR spectrum (see Figure S1 of the Supporting Information) showed, besides the two NEt$_4$ multiplets, one singlet at δ = 6.93. The $^{13}$C NMR spectrum (Figure S2) showed (besides the two NEt$_4$ carbon atoms) 10 signals, hinting to two different substances **A** and **B** of the type [C$_5$X(CN)$_4$] (three signals for the ring carbons and two for the cyano groups each). Comparison with the NMR data of Ag[C$_5$H(CN)$_4$], which we had prepared before on a different route [22], suggests that one of these compounds (**A**) is the tetraethylammonium salt of tetracyanocyclopentadienide, NEt$_4^+$ [C$_5$H(CN)$_4$]$^-$ (**3**). Our attempt to reproduce the synthesis of compound **2b** also yielded a product mixture. The $^1$H- NMR spectrum (Figure S3) looked very similar to the one obtained with NEt$_4$Cl, suggesting that **3** had also been formed. This interpretation was supported by the $^{13}$C NMR spectrum (Figure S4), which showed (besides the two NEt$_4$ signals) 15 signals, suggesting three different substances **A**, **B** and **C** all of the type [C$_5$X(CN)$_4$]. Two of the signal sets were identical to the ones obtained with NEt$_4$Cl, and therefore the formation of compound **3** can be assumed for this reaction as well. The identity of compound **B** remained unclear, as its formation in both reactions excluded the presence of a halide. Since recrystallization attempts did not turn out successful in both cases (but see Section 2.1.3), both products were treated with AgNO$_3$ in acetone.

From the original NEt$_4$Cl product, a mixture was obtained again which could be separated by column chromatography. A first fraction turned out to be the already known Ag[C$_5$H(CN)$_4$] (Figures S5 and S6), while the second was obviously the silver analogue **B′** of the second product of the NEt$_4$ starting material because of the similarity with the $^{13}$C NMR data (Figure S7). Recrystallization of this product from toluene/acetonitrile yielded X-ray quality crystals. The crystal structure determination identified the product **B′** as the complex Ag$_2$[C$_{10}$(CN)$_8$] (**4**) which contains the so-far unknown octacyanofulvalenediide dianion. It is therefore very likely that the unidentified product **B** is the NEt$_4$ analogue **5** of the silver complex **4**.

### 2.1.2. Synthesis of K$_2$[C$_{10}$(CN)$_8$] (**6**)

Salt metathesis of the silver complex **4** with KCl in MeOH yielded the corresponding potassium complex salt K$_2$[C$_{10}$(CN)$_8$] (**6**) as a colorless solid. Its purity was confirmed by $^1$H- and $^{13}$C-NMR spectra (Figures S8 and S9). Recrystallization allowed the isolation of X-ray quality crystals. Our results are summarized in Scheme 2.

### 2.1.3. Reaction of Diazotetracyanocyclopentadiene with [Ru(C$_5$H$_5$)Cl(PPh$_3$)$_2$]

It was mentioned in the Introduction that treatment of C$_5$X$_4$N$_2$ (X = Br, Cl) with a palladium complex yielded the corresponding octahalofulvalenes. We reasoned that compound **1** might also react with catalytically active metals. Just by chance, we chose the ruthenium complex [Ru(C$_5$H$_5$)Cl(PPh$_3$)$_2$] for our study. We studied the reaction of **1** both with stoichiometric or catalytic amounts of this ruthenium compound in MeCN as the solvent, using either reflux temperature (100 °C) or 0 °C. After addition of NEt$_4$Cl or NEt$_4$Br and chromatographic work-up, the main product was compound **3**. However, MS analysis (ESI or FAB) showed that octacyanofulvalenediide had also been formed, i.e., compound **5**. This was confirmed by crystal structure analysis.

### 2.2. Cyclovoltammetry

Cyclic voltammetry (CV) of crystalline **6** reveals one oxidation step at 1.25 V during oxidative scanning and a broad shoulder at 1.4 V in the reverse scan, which could be interpreted as partial reversibility similar to observations in several other tetracyanocyclopentadienides (Figure 1). Since **6** takes the form of a dianion in its initial stage, oxidation leads to a radical mono-anion. While the possibility of a two-electron transfer to a neutral compound (analogous to a fulvalene compound) cannot be excluded due to the broad shoulders of the voltagramm, it would not be supported by results from the preparative part of this work, which gave no indication of such a compound being present. The lack of

a neutral fulvalene analogue in CV might indicate that the neutral form of this compound is highly unfavorable, while the radical and dianionic forms are better stabilized. This would fit the electron-deficient nature of the system, which in theory should favor the charged states.

**Scheme 2.** Summary of the reactions described in Sections 2.1.1 and 2.1.2. The formation of **2a** was not observed in our study, but was reported in the literature.

### 2.3. EPR Spectroscopy

EPR measurements show signals in both the compounds **4** and **6** with g values of 2.001 and 2.002 when subjected to UV or sunlight, similar to measurements of other 1,2,3,4 tetracyanocyclopentadienide derivatives, indicating free radical formation by photoinduced electron transfer (Figure 2). While the signals of **6** are short-lived, **4** shows less degradation over time, a property also observed in other derivatives of this ligand class.

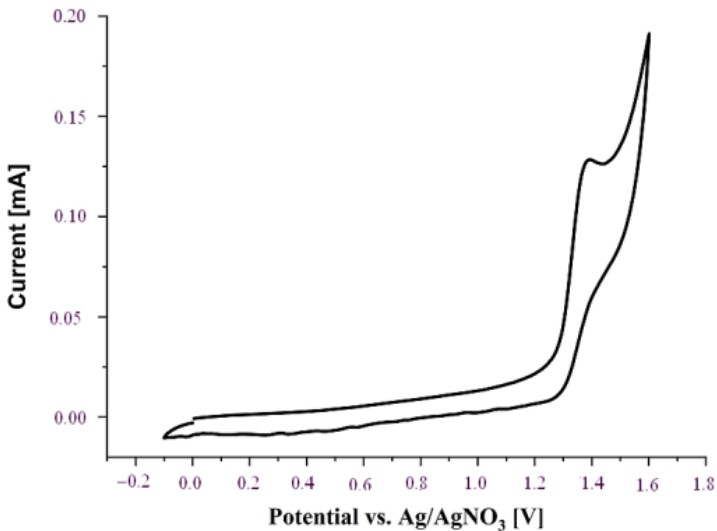

**Figure 1.** CV curve of crystalline **6** (0.1 mM) in MeCN at 25 °C (0.1M [NBu$_4$][PF$_6$] supporting electrolyte; scan rate 100 mV s$^{-1}$; Pt-wire as working electrode, Ag/AgNO$_3$ as reference electrode.

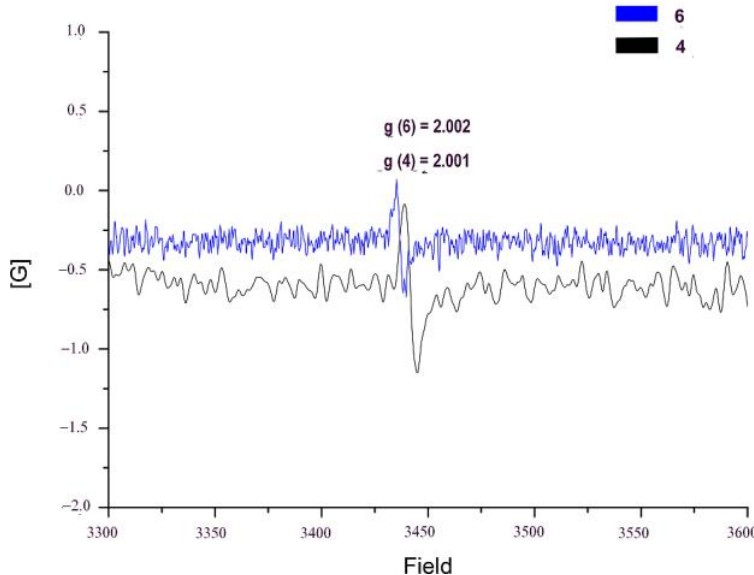

**Figure 2.** EPR spectra of compounds **4** and **6** after being subjected to sunlight.

### 2.4. Crystallography

2.4.1. Molecular and Crystal Structure of Compound **1**

Compound **1** crystallizes in the orthorhombic space group *Pbca* with one molecule in the asymmetric unit. Figure 3 shows an ORTEP3 view of the molecular unit. Table 1 collects some important metric parameters of the molecule. Although there is still some variation in the C–C bond lengths of the ring (two "short" bond lengths of ca. 1.393(1) Å and three "long" ones with ca. 1.418(1) Å), they are definitely more delocalized than in the structure of diazo-tetraphenylcyclopentadiene [23]. In this compound, the "short" bonds are at 1.373(3) Å and the "long" bonds average at 1.450(3) Å. In addition, the N–N bond of this compound is with 1.121(3) Å, which is substantially longer, while the C–N$_2$ bond is significantly shorter (1.316(3) Å). Unfortunately, there are no more diazocyclopentadiene structures available for comparison. However, there are some DFT calculations on the parent compound, which yield C–C bond lengths of 1.369 and 1.447 Å and a C–N$_2$ bond length of 1.312 Å [24].

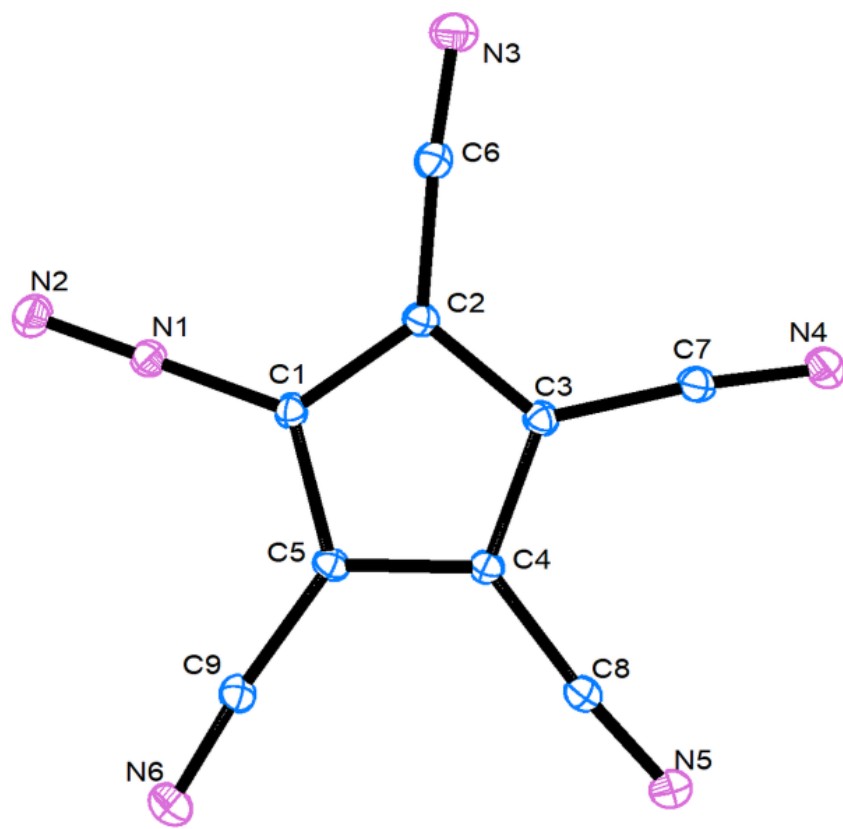

**Figure 3.** ORTEP3 view of the molecular unit of compound **1**.

**Table 1.** Important bond parameters of compound **1**.

| Bond | Length [Å]/Angle [°] | Bond | Length [Å]/Angle [°] |
|---|---|---|---|
| N1–N2 | 1.1017(13) | C1–N1 | 1.3565(13) |
| C1–C2 | 1.4202(14) | C2–C3 | 1.3919(14) |
| C3–C4 | 1.4164(14) | C4–C5 | 1.3937(14) |
| C5–C1 | 1.4212(14) | | |
| $C_{cp}$–$C_{CN}$ | 1.4225(15)–1.4252(15) | $C_{CN}$–$N_{CN}$ | 1.1475(15)–1.1502(15) |
| C1–N1–N2 | 179.7(1) | $C_{cp}$–C–N | 174.3(1)–175.4 [1] |

[1] $C_{cp}$ are the carbon atoms of the ring, while $C_{CN}$ are the carbon atoms of the nitrile groups.

When looking at intermolecular interactions, there are several short N...N contacts, particularly between atoms N1 and N5' (1/2 +*x*, 1.5–*y*, 1–*z*) which are closer by 0.23 Å than the sum of their Van der Waals radii. Thus, an infinite 1D chain forms with the base vector [1 0 0] (Figure S10).

2.4.2. Crystal and Molecular Structure of Compound **4**, Toluene Solvate

Compound **4** crystallizes as a bis-toluene solvate $Ag_2[C_{10}(CN)_8] \times 2(C_7H_8)$ in the monoclinic space group *C2/c* with half a molecule in the asymmetric unit. All crystals were twinned and were refined against a HKL5 dataset; in addition, the toluene molecules were disordered (50:50) across an inversion center. Two crystals were measured, one at 298 K and one at 100 K. Since the results of both refinements did not produce significant differences, here only the low temperature results are discussed. The results of the room-temperature determination are reported in the Supplementary Information.

Figure 4 shows an ORTEP3 view of the asymmetric unit (Figure S11 shows the corresponding plot of the r.t. determination), Figure 5 shows the coordination sphere of the unique silver atom and Figure 6 displays the anion with all coordinated silver ions.

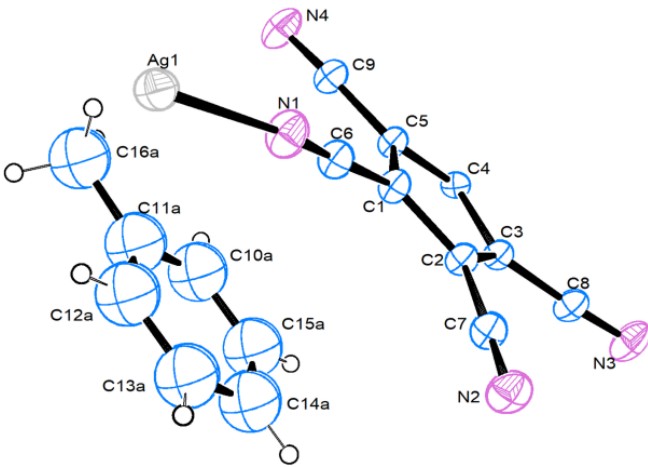

**Figure 4.** The asymmetric unit of compound **4** (only one orientation of the toluene solvent shown).

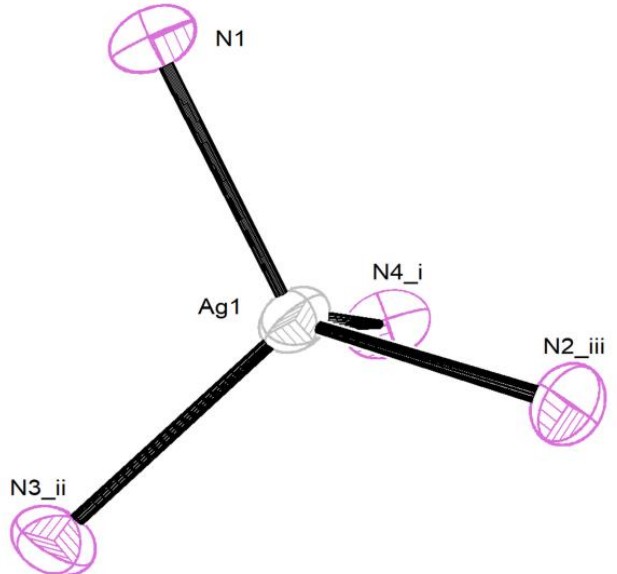

**Figure 5.** The coordination sphere of Ag1. Symmetry operators: i: 1/2–*x*, 1/2–*y*, 1–*z*; ii: *x*–1/2, 1.5–*y*, *z*–1/2; iii: 1/2–*x*, *y*–1/2, 1.5–*z*.

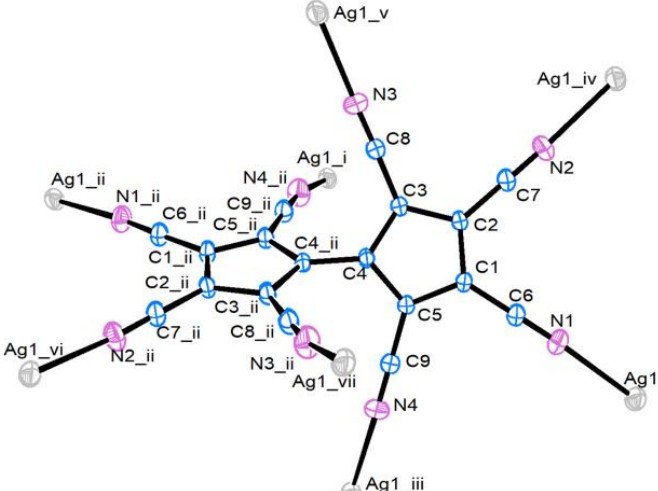

**Figure 6.** The octacyanofulvalenediide dianion with its eight coordinated Ag$^+$ ions.

The silver ion is tetrahedrally coordinated by four different octacyanofulvalenediide anions, while each dianion is coordinated to eight silver ions using all of its eight cyano nitrogen atoms. The two halves of the dianion are joined by a single bond between C4 and C4_ii (symm. op.: 1–$x$, $y$, 1.5–$z$) and are twisted by ca. 54° (which is an equivalent description of the torsion angle calculated as −126.1°). Other important bond parameters are collected in Table 2.

**Table 2.** Important bond parameters of compound **4** (low-temperature determination).

| Bond | Length [Å]/Angle[°] | Bond | Length [Å]/Angle [°] |
|---|---|---|---|
| Ag1–N1 | 2.248(4) | Ag1–N2_iii | 2.246(4) |
| Ag1–N3_ii | 2.301(5) | Ag1–N4_i | 2.309(5) |
| (C–C)$_{cp}$ | 1.399(6)–1.417(6) | C$_{cp}$–C$_{CN}$ | 1.412(6)–1.424(6) |
| (C–N) | 1.134(7)–1.147(7) | C4–C4_ii | 1.462(8) |
| C1–C6–N1 | 178.6(5) | C2–C7–N2 | 177.9(5) |
| C3–C8–N3 | 177.1(6) | C5–C9–N4 | 177.1(6) |
| Ag1–N1–C6 | 169.2(4) | Ag1–N2–C7 | 168.7(4) |
| Ag1–N3–C8 | 164.1(5) | Ag1–N5–C9 | 164.2(5) |
| C3–C4–C4_ii–C3_ii | −126.1(5) | | |

The symmetry operators given in the Ag–N distances refer to Figure 5, while the torsion angle refers to Figure 6.

Compound **4** forms a three-dimensional polymer structure, with base vectors [1 0 0], [0 1 0] and [0 0 1]. Figure 7 shows a packing plot watched along the crystallographic *b* axis, and Figure 8 a packing plot along *a*.

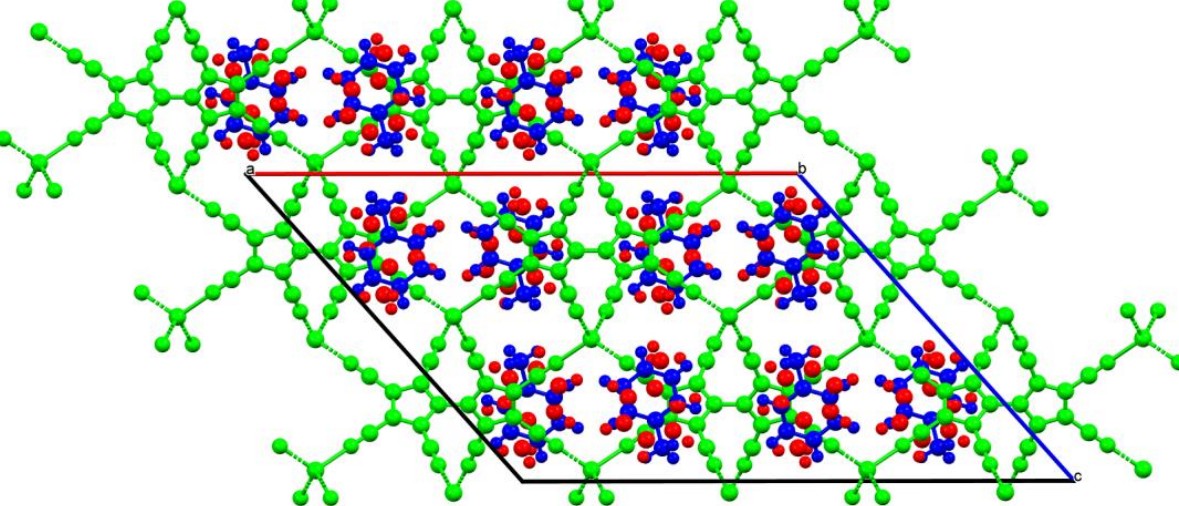

**Figure 7.** Packing plot (MERCURY) of compound **4**, watched along *b*. Color coding: green is the coordination polymer, while blue/red are the disordered toluene molecules.

Although we were not able to remove the toluene chemically, we just removed it virtually from the crystal structure. Its exclusion reveals a network of pores with a diameter of 5.8 Å at the bottleneck. The calculation of solvent accessible voids shows a volume of 1664 Å$^3$ per cell, corresponding to 56% of cell volume. The space takes the form of helical channels running parallel to the crystallographic *b* axis (Figure S12). At the same time, exclusion of the toluene guests from the packing diagrams shows that there exist a series of fused [Ag$_2$(NC–(C$_n$)–CN)$_2$] ring systems, as they are quite often observed in the structures of silver polycyanocyclopentadienides. Figure S13 shows the common 14-membered rings (involving nitrile nitrogens N2 and N3) and two different 20-membered rings involving either nitrogens N1 and N3 or N2 and N4. In addition, there are helices winding down the *b* axis involving nitrogens N1 and N2. Ag...Ag distances across the rings are 7.023 and 8.799 Å, respectively, and across the helix the distance is 7.363 Å.

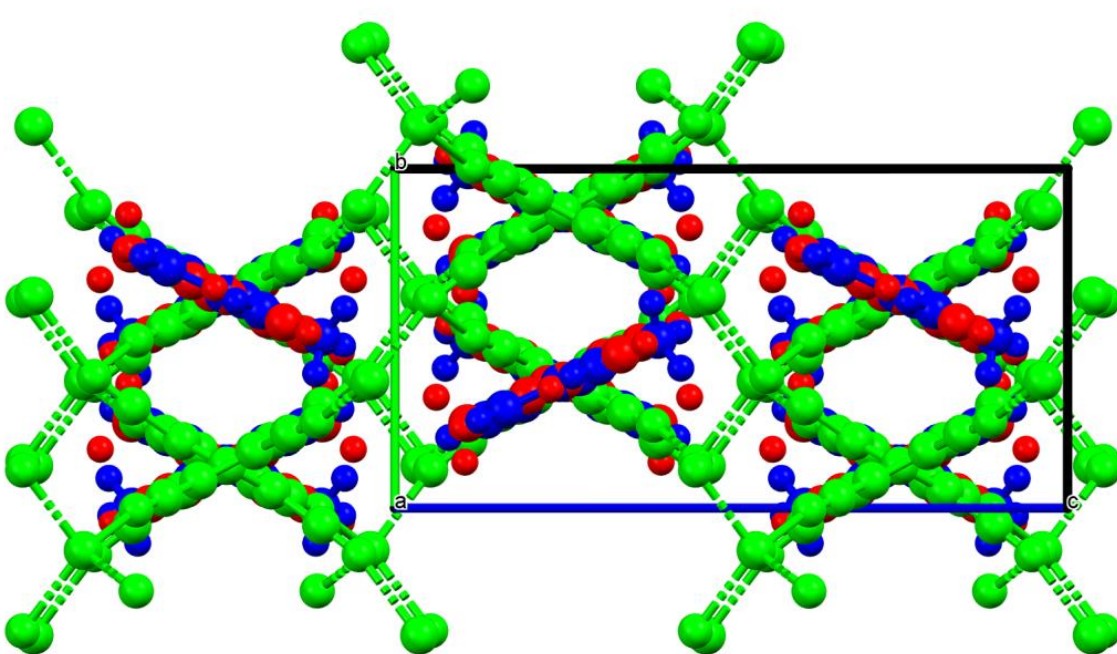

**Figure 8.** Packing plot (MERCURY) of compound **4**, watched along *a*. Same color coding as in Figure 7.

### 2.4.3. Crystal and Molecular Structure of Compound 5

Compound **5** crystallizes in the orthorhombic space group *Pbcn* with two independent half molecules in the unit cell. Figure 9 shows an ORTEP3 plot of the asymmetric unit.

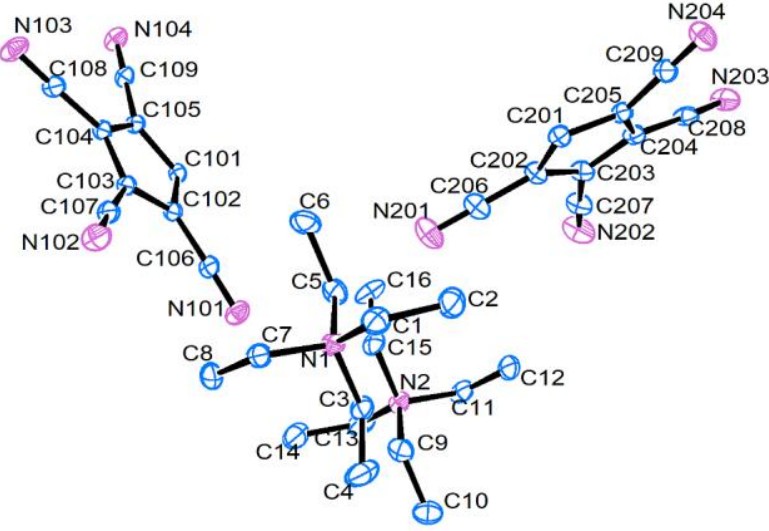

**Figure 9.** The asymmetric unit of compound **5**.

The half molecules are completed via single bonds between atoms C101–C101[i] and C201–C201[i], respectively, created by a twofold rotation axis through the middle of these bonds (symmetry operator: 1–*x*, *y*, 1.5–*z*). The complete octacyanofulvalenediide dianion of molecule A is shown in Figure 10. The two rings are twisted by ca. 54°. The bond parameters of the two independent molecules are very similar and are collected in Table 3.

The unit cell contains small cavities at the corners and the center, making up for approximately 1.1% of the volume. They are, however, too small to accommodate solvent molecules. As there is no metal ion in the structure, and no "classical" H-bond donor, all anions are isolated and separated from each other. Figure 11 shows a packing plot of this structure.

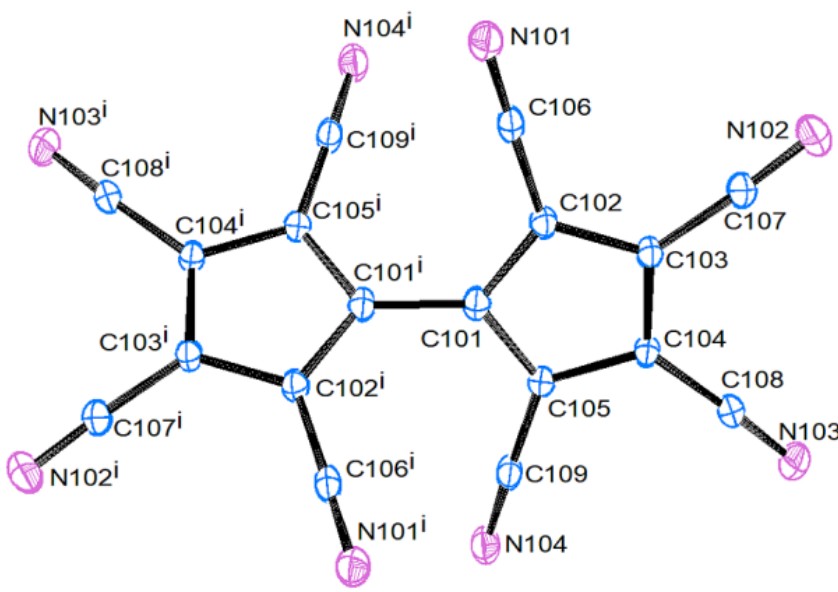

**Figure 10.** The complete octacyanofulvalenediide dianion of molecule A of compound **5**.

**Table 3.** Bond parameters of compound **5**.

| Bond | Length [Å]/Angle [°] | |
|---|---|---|
| | Molecule A | Molecule B |
| $(C–C)_{cp}$ | 1.399(3)–1.418(3) | 1.397(3)–1.428(3) |
| $(C–N)$ | 1.145(3)–1.152(3) | 1.143(3)–1.151(3) |
| $C_{cp}–C_{CN}$ | 1.418(3)–1.428(3) | 1.417(3)–1.429(3) |
| $Cn01–Cn01^i$ | 1.467(4) | 1.472(4) |
| $Cn02–Cn06–Nn01$ | 178.3(3) | 179.5(3) |
| $Cn03–Cn07–Nn02$ | 176.4(3) | 177.7(3) |
| $Cn04–Cn08–Nn03$ | 176.8(3) | 177.5(3) |
| $Cn05–Cn09–Nn04$ | 178.8(3) | 179.2(3) |
| $Cn02–Cn01–Cn01^i–Cn02^i$ | −125.9(3) | −54.9(3) |

$(C–C)_{cp}$ are the C–C bond lengths within one cyclopentadienyl ring; $C_{cp}$ and $C_{CN}$ are the individual carbon atoms of the cyclopentadienyl ring and the attached carbon atoms of the nitrile groups; $Cn0x$ (n = 1 or 2, x = 1–9) are the corresponding C atoms of molecules A and B.

2.4.4. Crystal and Molecular Structure of $[K(H_2O)]_2[C_{10}(CN)_8]$, **6a**

The aqua complex **6a** crystallizes in the orthorhombic space group *Ccca* with a quarter molecule in the asymmetric unit (Figure 12).

The potassium ions are coordinated by six nitrile nitrogens (from five different dianions) and two water oxygens (Figure 13). Pairs of K⁺ ions related by [*x*, *y*, *z*] and [−*x*−1/2, −*y*, *z*] form zig-zag chains of face-fused (distorted) octahedra along the *x* direction, with metal–metal distances of 3.99 Å (Figure S14). The dianion is coordinated to ten K⁺ ions, of which two are bridging two nitrile functions of the same dianion (Figure 14). Important bond parameters are collected in Table 4.

Compound **6a** forms a 3D polymer with base vectors [1 0 0], [0 1 0] and [0 0 1]. The unit cell contains ca. 3% solvent-accessible voids. The coordinated water molecules form weak hydrogen bonds to nitrile nitrogens N2. A packing plot is displayed in Figure 15.

Pairs of independent zig-zag chains of potassium ions (viewed in superposition in Figure 15, or displayed separately in Figure S15), as also shown for a single chain in Figure S14, run along the *a* axis at *y* = 0 and *y* = 0.5, and are bridged by octacyanofulvalenediides via nitrogens N1 in the *b* direction and N2 in the *c* direction (Figure S16). The closest approach of two K⁺ ions in the *b* direction is 8.756 Å, while in the direction of the *ab* diagonal it is 7.601 Å.

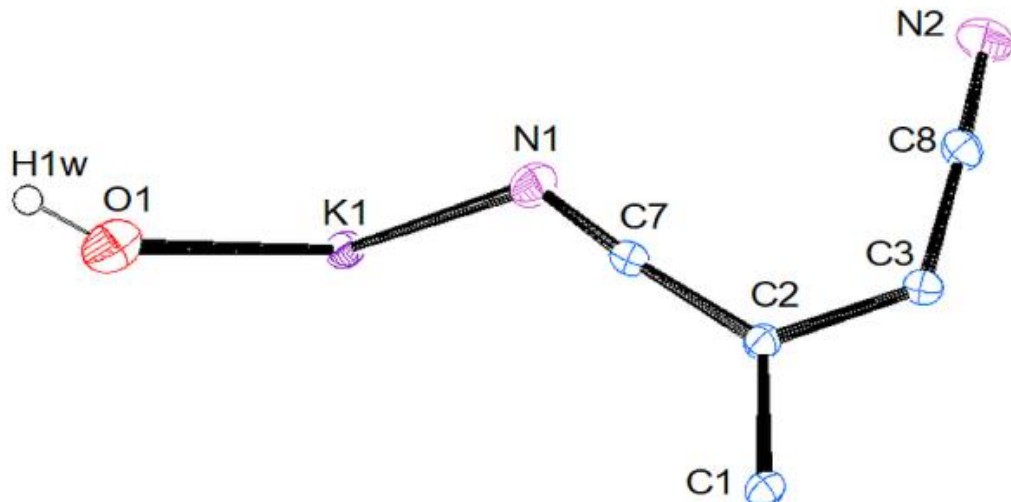

**Figure 11.** Packing plot (MERCURY) of compound **5**, watched along the crystallographic *c* axis. Color coding: blue and yellow are the NEt$_4$ cations, yellow and red the [C$_{10}$(CN)$_8$] dianions.

**Figure 12.** ORTEP3 view of the asymmetric unit of **6a**.

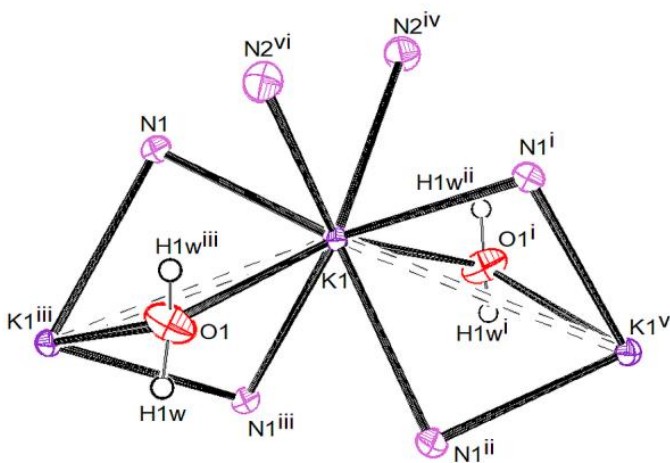

**Figure 13.** The coordination sphere of K$^+$ in compound **6a**. Symmetry operators: i: –1–*x*, *y*, 1/2–*z*; ii: *x*–1/2–*y*, 1/2–*z*; iii: –*x*–1/2, –*y*, *z*; iv: *x*–1/2, *y*, –*z*; v: –*x*–1.5, –*y*, *z*; vi –*x*–1.5, *y*, 1/2–*z*.

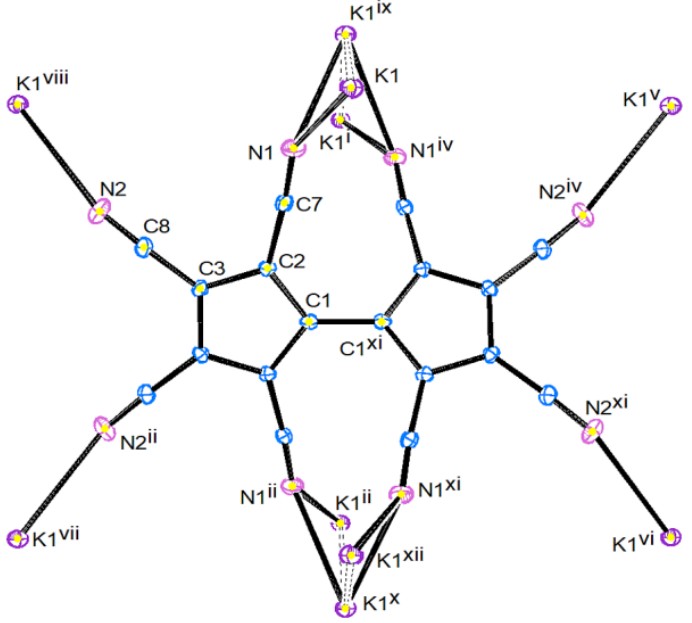

**Figure 14.** A complete octacyanofulvalenediide dianion with all coordinated K$^+$ ions. Symmetry operators: i: 1+*x*, *y*, *z*; ii: –*x*, –*y*–1/2, *z*; iv: –*x*, *y*, 1/2–*z*; v: 1/2+*x*, *y*, 1–*z*; vi: –*x*–1/2, –*y*–1/2, 1–*z*; vii: –*x*–1/2, –*y*–1/2, –*z*; viii: 1/2+*x*, *y*, –*z*; ix: –*x*–1/2, –*y*, *z*; x: 1/2+*x*, *y*–1/2, *z*; xi: *x*, –*y*–1/2, *z*; xii: –1–*x*, –*y*–1/2, *z*.

**Table 4.** Important bond parameters of compound **6a**.

| Bond | Length [Å]/Angle[°] | Bond | Length [Å]/Angle [°] |
|---|---|---|---|
| K1–N1/K1–N1$^i$ | 2.839(2) | K1–N2$^{iv}$/K1–N2$^{vi}$ | 2.870(2) |
| K1–O1 | 2.874(2) | K1–N1$^{ii}$/K1–N1$^{iii}$ | 2.915(2) |
| K1–K1$^{iii}$ | 3.9929(6) | | |
| (C–C)$_{cp}$ | 1.400(4)–1.418(3) | C$_{cp}$–C$_{CN}$ | 1.418(3)–1.426(3) |
| (C–N) | 1.146(3)–1.147(3) | C1–C1$^{xi}$ | 1.477(6) |
| C2–C7–N1 | 175.3(2) | C3–C8–N2 | 176.8(3) |
| K1–N1–C7 | 132.3(2) | K1–N1$^{ii}$–C7$^{ii}$ | 131.6(2) |
| K1–N2–C8 | 150.0(2) | | |
| C2–C1–C1$^{xi}$–C2$^{xi}$ | −140.3(2) | | |

The symmetry operators in the distances and angles involving K1 refer to Figure 13, while the ones involving only the anion refer to Figure 14.

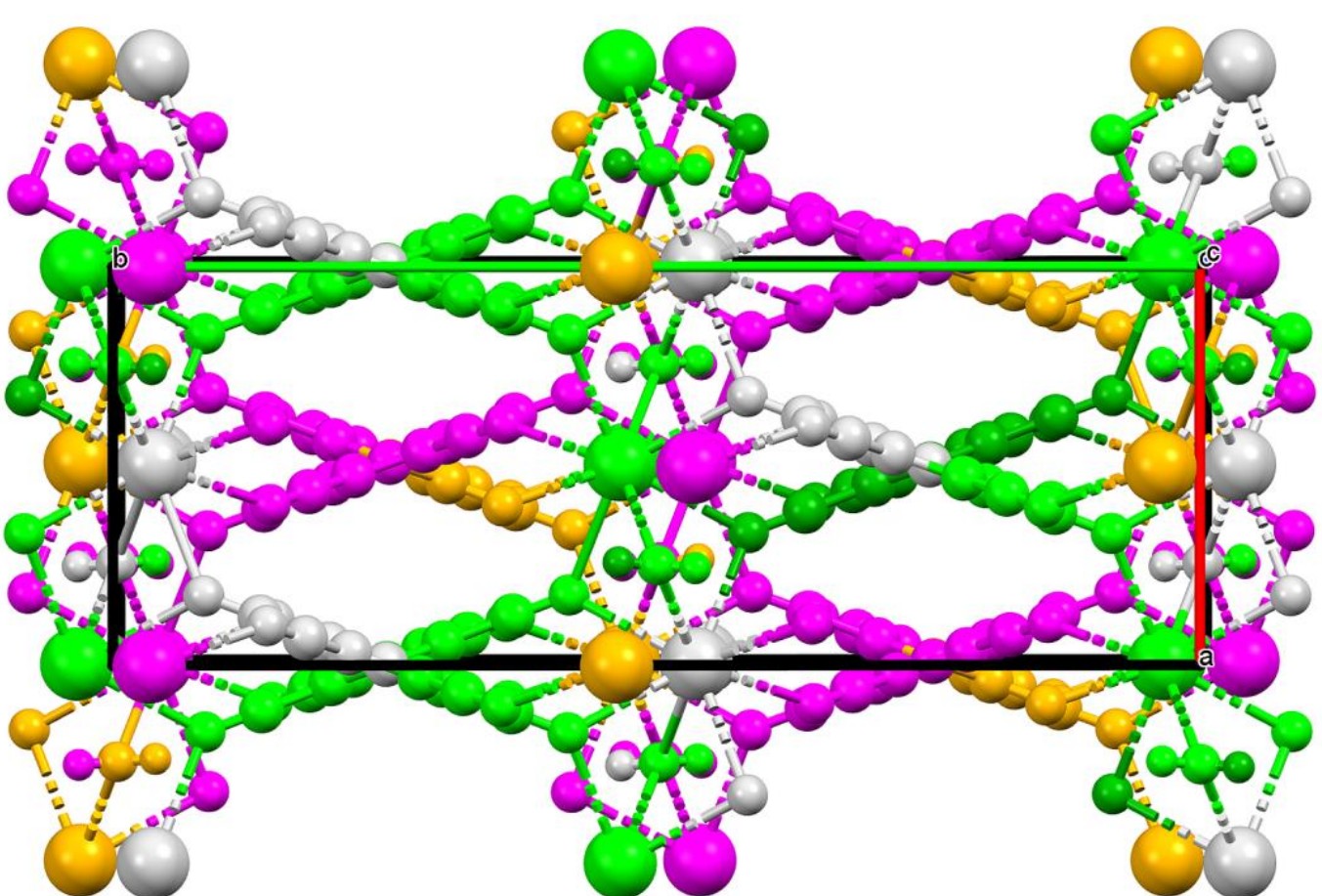

**Figure 15.** Packing plot of compound **6a**, watched along the crystallographic *c* axis. Color coding "by symmetry operation" (default settings of MERCURY).

## 3. Discussion

Diazotetracyanocyclopentadiene had been reported to behave as an aromatic diazonium ion, based on its reactivity [1] and its $^{13}$C- and $^{15}$N-NMR data [25]. Our crystal structure determination supports this view. Its bond length alternation parameter $\Delta_R$ is only 0.018 Å, much smaller than the 0.078 Å calculated for $C_5H_4N_2$, which was already regarded as small and as having a strong indication of aromaticity [24]. We also performed some DFT calculations (B3LYP/6-311G(dp) level) using the program *CrystalExplorer* [26]. Figures S17–S19 show HOMO/LUMO contours, an electron density visualization and two views of the electrostatic potential. In addition, these results support the high aromaticity of the cyclopentadienyl unit. With this background, the typical reactivity of arenediazonium ions towards metals and metal salts, i.e., radical reactions, should be expected for compound **1**. Unfortunately, the major reaction of these radicals is H abstraction to give the $[C_5(CN)_4H]$ anion. The usual main products of the Cu(I)-catalyzed coupling of aromatic diazonium ions, i.e., biaryls and azo compounds [27,28], were formed only in small yields or not at all. (It should be mentioned, however, that the original publication on the synthesis and reactivity of compound **1** reports the formation of an azo compound in the thermal reaction of **1** with CuCN in MeCN [1].) Although our study was not a true mechanistic one, we assume that the low yields of coupling products are the consequence of both steric and electronic repulsions: the two "ortho" cyano groups and the negative charges inhibit the mutual approach of the two radical anions, and thus the H abstractions dominate the reaction kinetics. The fact that rather high yields of coupling products can be obtained from the reaction of **1** with [RuCp(PPh$_3$)$_2$Cl] remains rather mysterious. The only comparable reaction in the literature is the reaction of [RuCp*(P(OR)$_3$)$_2$Cl] with 1,1-diaryldiazomethanes,

which was reported to give cationic N-coordinated Ru diazoalkane complexes [29] with no indication of any coupling products. This result is reminiscent of the N-coordinated $C_5Cl_4N_2$ complexes mentioned above [18]. Now, the only "explanation" might be that the aromaticity of compound **1** drives the reaction into another direction compared with "true" diazoalkanes and less aromatic diazocyclopentadienes.

The structure of compound **5** can be regarded as the structure of the "free" octa-cyanofulvalenediide anion because there are no coordinating cations or strong H-bond donors (there are, however, very weak interactions with aliphatic C–H bonds, but their influence on bond parameters can be neglected). The cyclopentadienyl rings are aromatic, as can be derived from the bond alteration parameter $\Delta_R$, which amounts to 0.019 Å in molecule A and 0.031 Å in molecule B. In addition, the bonds to the nitrile carbon atoms have the same lengths (within $2\sigma$) as the bonds within the ring. The central C–C bond between the two cyclopentadienyl halves is a typical single bond with ca. 1.47 Å. The two cyclopentadienyl rings are twisted by ca. 55°. In the $Ag^+$ complex **4** (low temperature), these bond parameters hardly change. $\Delta_R$ amounts to 0.018 Å and the bonds within the ring and between the ring and nitrile carbons are identical within $2\sigma$. Both halves of the fulvalenediide anion are bound via a single bond and are twisted by ca. 54°. In the $K^+$ complex **6**, the bond alteration parameter also amounts to 0.018 Å. The exocyclic C–C bonds are slightly longer than the endocyclic ones. The single bond between the two cyclopentadienyl rings is ca. 1.48 Å long, with the two halves being twisted by ca. 40°. This significantly smaller twist angle is a consequence of the fact that the potassium ion bridges two of the "ortho" nitriles.

The different sizes of $Ag^+$ ("effective ionic radius" 1.00 Å for CN 4, [30]) and $K^+$ (1.51 Å for CN 8, [30]) lead to significant differences in the coordination spheres. While the $Ag^+$ ion is tetrahedrally coordinated by four nitrile nitrogens with bond distances ranging from 2.25–2.31 Å, the $K^+$ ion has coordination number 8, $KN_6O_2$ polyhedra, with bond distances between 2.84 and 2.92 Å. The Ag–N–C bonds deviate only slightly (11–16°) from linearity, but the K–N–C bonds can be regarded as "bent", particularly at atoms N1. Both structures might be compared with the structures of $[AgC_5(CN)_5]$ and $[KC_5(CN)_5]$, respectively [31,32]. These compounds show different coordination polyhedra dependent on the solvent the crystals came from. Crystals of the $Ag^+$ salt from EtOH show severely distorted $AgN_4$ tetrahedra, with Ag–N distances ranging from 2.236(5) to 2.378(6) Å, N–Ag–N angles between 88.0(2) and 129.7(2)°, and Ag–N–C angles between 149.7(6) and 169.9(5)°. The pentacyanocyclopentadienide anion uses only four of its nitrile groups for coordination. The compound forms an interwoven 3D network with very large pores, similar to the situation found in **4** [31]. $[KC_5(CN)_5]$, when grown from isopropanol/pentane, shows an octahedral metal ion with K–N bonds ranging from 2.777(1) to 2.9537(6) Å and a pentacyanocyclopentadienide using all five nitrile groups for coordination, with one bridging two $K^+$ ions in 4.975 Å distance, and K–N–C angles ranging from 122.1 to 138.0° [32]. Comparison of these compounds with our compounds **4** and **6** shows a wider range of metal-nitrogen distances in the $[C_5(CN)_5]$ salts and also a higher degree of bending in the M–N–C groups. Alternatively, one might use for comparison the structures of the tricyanomethanides $M[C(CN)_3]$. While the silver salt has a coordination number of three with Ag–N distances at 2.16 and 2.27 Å and Ag–N–C angles of 172.8° and 153.7° [33], the potassium salt has seven nitrile nitrogens coordinated at 2.86–2.98 Å and K–N–C angles ranging from 114 to 159° [34]. The $K[C(CN)_3]$ contains triply N-bridged $K^+$ ions with a K...K distance of 3.89 Å.

Octacyanofulvalenediide uses, in both compounds **4** and **6,** all of its eight nitrile groups for coordination; in compound **6,** four of them coordinate to two symmetry-related $K^+$ ions each, which is similar to the situation found in $[KC_5(CN)_5]$. Therefore, both compounds form three-dimensional coordination polymers, with the potassium structure being more "complicated".

## 4. Materials and Methods

### 4.1. Starting Materials and Instrumentation

Reagents (Cu, NEt$_4$Cl, NEt$_4$Br, AgNO$_3$, [Ru(C$_5$H$_5$)Cl(PPh$_3$)$_2$]) and solvents (MeCN, acetone, toluene) were obtained commercially in analytical grade and used as obtained, except for anhydrous MeCN, which was saturated with argon. Diazotetracyanocyclopentadiene (**1**) was prepared according to the literature [1].

CV measurements were performed with an Autolab potentiostat/galvanostat (PG-STAT302N) with a FRA32M module operated with Nova 1.11 software and a conventional three-electrode setup. Two platinum wires were used as the working and counter electrode, respectively. A Ag/0.01M AgNO$_3$ electrode was utilized as reference electrode.

### 4.2. Reaction of **1** with Cu and NEt$_4$Cl in MeCN

A suspension of powdered copper (0.57 g, 9.0 mmol) and NEt$_4$Cl (1.58 g, 9.5 mmol) in acetonitrile (30 mL) was heated to 90 °C. A solution of **1** (1.13 g, 5.9 mmol) in MeCN (19 mL) was added to the boiling mixture within one hour. After cooling down to room temperature and filtering, the solvent was removed in vacuo. A brown powder was obtained (2.18 g). $^1$H- and $^{13}$C{$^1$H}-NMR spectra (Figures S1 and S2) suggested the presence of two compounds: **A** (=**3**) and **B** (=**5**).

Compound **3**: $^1$H NMR (400 MHz, DMSO-d$^6$): $\delta$= 6.93 (s, C$_5$H), 3.19 (q, NCH$_2$), 1.15 (tt, CH$_3$). $^{13}$C-NMR (101 MHz, DMSO-d$^6$): $\delta$ = 123.9 (CH), 116.0, 114.8 (2 × CN), 99.4, 96.3 (2 × CCN), 51.3 (t, NCH$_2$), 7.0 (s, CH$_3$). MS: FAB(+): *m/z* = 130.2 (NEt$_4^+$); FAB(−): *m/z* = 165.3 ([C$_9$HN$_4^-$]).

Compound **5**: $^1$H NMR (400 MHz, DMSO-d$^6$): $\delta$= 3.19 (q, NCH$_2$), 1.15 (tt, CH$_3$). $^{13}$C-NMR (101 MHz, DMSO-d$^6$): $\delta$ = 127.5 (CH), 115.1, 114.5 (2 × CN), 99.4, 96.3 (2 × CCN), 51.3 (t, NCH$_2$), 7.0 (s, CH$_3$). MS: FAB(+): *m/z* = 130.2 (NEt$_4^+$); FAB(−): *m/z* = 164.2 ([C$_{18}$N$_8^{2-}$]).

### 4.3. Reaction of **1** with NEt$_4$Br in MeCN

A solution of **1** (3.00 g, 15.6 mmol) and NEt$_4$Br (10.10 g, 48.1 mmol) in MeCN (160 mL) was heated at 50 °C for 30 min. Then, the solvent was evaporated in vacuo, and water (150 mL) was added and the resulting suspension was filtered. The residue on the filter was washed with water (50 mL) and dried in vacuo. A brown powder was obtained (3.06 g). $^1$H- and $^{13}$C{$^1$H}-NMR spectra (Figures S3 and S4) showed the formation of three compounds: **A** (=**3**), **B** (=**5**) and **C** (=**2b**).

Compound **2b**: $^{13}$C-NMR (101 MHz, DMSO-d$^6$): $\delta$ = 114.9, 114.0 (2 × CN), 107.2 (CBr), 100.0, 98.4 (2 × CCN), 51.3 (t, NCH$_2$), 7.0 (s, CH$_3$). MS: FAB(+): *m/z* = 130.2 (NEt$_4^+$); FAB(−): *m/z* = 243.1/ 245.1 ([C$_9$BrN$_4^-$])

### 4.4. Reaction of the Crude Product from Section 4.2 with AgNO$_3$ in Acetone–Water

A solution of the product mixture obtained in 4.2. (1.53 g) in acetone (55 mL) was treated with a solution of AgNO$_3$ (2.69 g, 15.8 mmol) in water (15 mL). Stirring was continued at r.t. with exclusion of light for 4 days. After removal of the solvents in vacuo, the residue was taken up in the minimum amount of MeCN and placed on top of a silica gel column. MeCN: toluene 3:7 eluted a first fraction, while elution with MeCN: toluene 1:1 produced a second fraction. After evaporation to dryness, yellow-brown powders were obtained. Both fractions were examined by NMR spectroscopy. The first turned out to be Ag[C$_5$H(CN)$_4$], Figures S5 and S6, while the second was identified as **B'** (=**4**) (Figure S7). Recrystallization of the second fraction from toluene/acetonitrile produced X-ray quality crystals.

Compound **4**: $^{13}$C-NMR (101 MHz, DMSO-d$^6$): $\delta$ = 128.0 (C–C), 115.2, 114.6 (2 × CN), 99.9, 96.9 (2 × CCN), MS: MALDI(-): *m/z* = 329.1 ([HC$_{18}$N$_8^-$]). IR: [cm$^{-1}$]: $\nu$ = 2218 vs, 1465 m, 1437 m, 730 vs, 695 m (+ many weak and very weak absorptions between 1900 and 750); EA: calc. for Ag$_2$C$_{18}$N$_8$ × 1.68 toluene × MeCN: C 51.57, N 17.03 H 2.24; found: C 51.54, N 16.97 H 2.29.

### 4.5. Reaction of **1** with Catalytic Amounts of [Ru(C₅H₅)Cl(PPh₃)₂] and NEt₄Br

A solution of **1** (2.0 g, 10.1 mmol) in anhydrous MeCN (50 mL) was added to a suspension of [Ru(C₅H₅)Cl(PPh₃)₂] (0.030 g, 4.1 µmol) in anhydrous MeCN (5 mL) at 0 °C within two hours. Then, water (20 mL) was added first and then solid NEt₄Br (6.00 g, 28.8 mmol). The obtained suspension was filtered, and the residue taken up in the minimum amount of MeCN. Chromatography on silica gel using MeCN: toluene 1:9 as eluent produced two major fractions. NMR spectroscopy identified the first fraction as NEt₄[C₅H(CN)₄]. The second fraction produced after evaporation a brown powder (0.40 g) which contained, according to its mass spectra, compound **5**. Recrystallization from MeCN/toluene produced a small number of crystals of low quality (pseudomerohedral twins).

### 4.6. Reaction of **4** with KCl in MeOH

A suspension of **4** (68 mg, 0.125 mmol) in methanol (10 mL) was treated with a solution of KCl (4.7 mg, 0.063 mmol) in methanol (5 mL). The mixture was stirred at room temperature under the exclusion of light for 8 h and the solvent was evaporated to a volume of 1 mL. The solution was purified in a syringe filter before the product was precipitated with dichloromethane as a colorless solid (23 mg). NMR spectra showed the presence of small amounts of a [C₅H(CN)₄] salt besides an octacyanofulvalenediide (presumably with K⁺ as cation, Figures S8 and S9). The compound was recrystallized from MeCN/toluene by vapor diffusion at 25 °C to yield X-ray quality crystals that were also subjected to further characterization.

Compound **6**: $^{13}$C-NMR (101 MHz, DMSO-d$^6$): $\delta$ = 128.0 (C–C), 114.6, 113.9 (2 × CN), 99.9, 96.9 (2 × CCN). MS (MALDI$^*$): $m/z$ = 328.1 (C₁₈N₈$^-$). calcd. for K₂C₁₈N₈ × 0.49 toluene × 2 H₂O:: C 52.79, N 22.98, H 1.64; found C 52.26, N 23.00, H 2.21.

### 4.7. Crystal Structure Determinations

All crystals were measured on a BRUKER D8VENTURE system. Compound **4** was obtained as twins. Refinement was possible using a HKLF 5 file with a BASF factor of ca. 0.25. Compound **5** also showed twinning (pseudomerohedral), which could, however, not be properly resolved. Still, refinement was possible. The experimental details of the structure determinations are collected in Table 5. The software package WINGX [35] was used for structure solution (SHELXT, [36]), refinement (SHELXL 2018/3, [37]), evaluation (PLATON, [38]), and graphical representation (ORTEP3 and MERCURY [35]). Carbon-bound hydrogen atoms were treated with a riding model using the AFIX command of SHELXL.

**Table 5.** Experimental details of the crystal structure determinations.

| Compound | 1 | 4-LT | 4-RT | 5 | 6 |
|---|---|---|---|---|---|
| Empirical formula | $C_9N_6$ | $C_{16}H_8AgN_4$ | $C_{16}H_8AgN_4$ | $C_{34}H_{40}N_{10}$ | $C_9H_2KN_4O$ |
| Formula weight | 192.15 | 364.13 | 364.13 | 588.76 | 221.25 |
| Temperature [K] | 100(2) | 100(2) | 297(2) | 100(2) | 110(2) |
| Crystal system | Orthorhombic | Monoclinic | Monoclinic | Orthorhombic | Orthorhombic |
| Space group | *P bca* | *C 2/c* | *C 2/c* | *P bcn* | *C cca* |
| Unit cell dimensions | | | | | |
| *a* [Å] | 12.5735(4) | 26.679(1) | 26.715(3) | 17.8082(6) | 7.4600(7) |
| *b* | 9.8840(3) | 7.3110(4) | 7.4955(7) | 17.7182(6) | 20.3621(19) |
| *c* | 13.6977(4) | 20.031(1) | 19.993(2) | 22.0972(7) | 13.2449(12) |
| *ß* [°] | | 131.793(1). | 131.929(3) | | |
| Volume [Å$^3$] | 1702.30(9) | 2912.9(3) | 2978.5(5) | 6972.3(4) | 2011.9(3) |
| Z | 8 | 8 | 8 | 8 | 8 |
| $\rho_{calc}$ [g cm$^{-3}$] | 1.499 | 1.661 | 1.624 | 1.122 | 1.461 |
| $\mu$ [mm$^{-1}$] | 0.104 | 1.381 | 1.350 | 0.070 | 0.503 |
| F(000) | 768 | 1432 | 1432 | 2512 | 888 |
| Crystal size [mm$^3$] | 0.060 × 0.050 × 0.040 | 0.090 × 0.060 × 0.020 | 0.090 × 0.060 × 0.020 | 0.100 × 0.080 × 0.050 | 0.100 × 0.030 × 0.010 |
| Θ range | 3.240–28.273° | 2.046–28.303° | 2.049–30.494° | 2.287–26.387° | 3.290–26.372° |
| Index ranges | −16 ≤ h ≤ 15, −11 ≤ k ≤ 13, −18 ≤ l ≤ 18 | −35 ≤ h ≤ 26, 0 ≤ k ≤ 9, 0 ≤ l ≤ 26 | −38 ≤ h ≤ 28, 0 ≤ k ≤ 10, 0 ≤ l ≤ 28 | −22 ≤ h ≤ 21, −21 ≤ k ≤ 22, −27 ≤ l ≤ 27 | −8 ≤ h ≤ 9, −25 ≤ k ≤ 20, −16 ≤ l ≤ 16 |
| Refl. coll. | 18615 | 3608 | 6594 | 68882 | 8675 |
| Indep. Refl. [R$_{int}$] | 2108 [0.0452] | 3608 [0.0440] | 6594 [0.0347] | 7143 [0.0708] | 1036 [0.0617] |
| Absorpt. correction | | | Semi-empirical from equivalents | | |
| T$_{max}$/T$_{min}$ | 0.7457/0.6852 | 0.7457/0.6621 | 0.6478/0.5642 | 0.9705/0.8778 | 0.9281/0.7917 |
| Data/restr./param. | 2108/0/136 | 3608/7/148 | 6594/7/148 | 7142/0/405 | 1036/0/70 |
| GOOF | 1.027 | 1.108 | 1.046 | 1.144 | 1.070 |
| R1/wR2 [I>2σ (I)] | 0.0365/0.1077 | 0.0445/0.1173 | 0.0580/0.1590 | 0.0626/0.1401 | 0.0413/0.0886 |
| R1/wR2 (all data) | 0.0494/0.1215 | 0.0508/0.1221 | 0.0742/0.1709 | 0.0933/0.1537 | 0.0563/0.0942 |
| Δ$\rho_{el}$ [e Å$^{-3}$] | 0.351/−0.225 | 1.571/−0.904 | 2.171/−1.112 | 0.243/−0.216 | 0.307/−0.291 |
| CCDC-# | 2236241 | 2236239 | 2236240 | 2236242 | 2236238 |

**Supplementary Materials:** The following supporting information can be downloaded at https://www.mdpi.com/article/10.3390/inorganics11020071/s1, Table S1: Important bond parameters of compound **4** (room-temperature determination); Figure S1: $^1$H-NMR (400 MHz, DMSO-d$^6$) of the crude reaction product of reaction 4.2; Figure S2: $^{13}$C{$^1$H}-NMR (101 MHz, DMSO-d$^6$) of the crude reaction product of reaction 4.2.; Figure S3: $^1$H-NMR (400 MHz, DMSO-d$^6$) of the crude reaction product of reaction 4.3; Figure S4: $^{13}$C{$^1$H}-NMR (101 MHz, DMSO-d$^6$) of the crude reaction product of reaction 4.3; Figure S5: $^1$H-NMR (400 MHz, DMSO-d$^6$) of the first chromatography fraction of reaction 4.4.; Figure S6: $^{13}$C{$^1$H}-NMR (101 MHz, DMSO-d$^6$) of the first chromatography fraction of reaction 4.4; Figure S7: $^{13}$C{$^1$H}-NMR (101 MHz, DMSO-d$^6$) of the second chromatography fraction of reaction 4.4; Figure S8: $^1$H-NMR (400 MHz, DMSO-d$^6$) of the purified product of reaction 4.6; Figure S9: $^{13}$C{$^1$H}-NMR spectrum (101 MHz, DMSO-d$^6$) of the purified product of reaction 4.6; Figure S10: The infinite 1D chain in the structure of compound **1**; Figure S11: ORTEP3 plot of the asymmetric unit of **4**, at r.t; Figure S12: Two views of the pore structure of **4**, generated by virtual removing the disordered toluenes; Figure S13: Two packing views of compound **4**; Figure S14: ORTEP3 view of the zig-zag chain of face-fused KN$_4$O$_2$ octahedra along the *x* direction; Figure S15: Packing plots of compound **6**, watched along *c*; Figure S16: Packing plot of compound **6**, watched along *b*; Figure S17: HOMO and LUMO of compound **1**, as calculated with *CrystalExplorer*; Figure S18: Electron density plot of compound **1**; Figure S19: Two views of the electrostatic potential distribution in compound **1**.

**Author Contributions:** Conceptualization, P.N. and K.S.; methodology, P.N.; software, K.S.; validation, P.N., and K.S.; formal analysis, P.N.; investigation, P.N. and Y.K.; resources, K.S.; data curation, K.S.; writing—original draft preparation, P.N. and Y.K.; writing—review and editing, K.S.; visualization, K.S. and P.N.; supervision, K.S.; project administration, K.S.; funding acquisition, Y.K. All authors have read and agreed to the published version of the manuscript.

**Funding:** This research received no external funding.

**Data Availability Statement:** CCDC-2236238-2236242 contain the supplementary crystallographic data for this paper. These data can be obtained free of charge via www.ccdc.cam.ac.uk/data_request/cif, by emailing data_request@ccdc.cam.ac.uk, or by contacting The Cambridge Crystallographic Data Centre, 12, Union Road, Cambridge CB2 1EZ, UK; fax: +44-1223336033.

**Acknowledgments:** We thank T. Klapötke for providing the NMR and D. Fattakhova-Rohlfing for providing the CV instrumentation. We also acknowledge P. Mayer for performing the X-ray diffraction measurements and F. Zoller for performing the CV experiments.

**Conflicts of Interest:** The authors declare no conflict of interest.

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
