# Peer review of "Coordination Chemistry of Polynitriles, Part XII—Serendipitous Synthesis of the Octacyanofulvalenediide Dianion and Study of Its Coordination Chemistry with K+ and Ag+"

_inorganics, doi:10.3390/inorganics11020071_

Round 1

Reviewer 1 Report

The article "Coordination Chemistry of Polynitriles, Part XII. Serendipitous Synthesis of the Octacyanofulvalenediide dianion and Study of  its Coordination Chemistry with K+ and Ag+" reported by Patrick R. Nimax et. al. is a nice piece of work and surtainely  enrich the area of coordination chemistry research and is worth for general reader. I recommend to publish this article in "Inorganics" subject to following minor revision to be address in revised manuscript-

1. There are some typo errors in the current version. Therefore, it is suggested that the authors thoroughly review the entire manuscript.

2. FT-IR, thermal study (TG/DTA) and PXRD analysis are missing in the manuscript. I think these studies will support the establishment of compounds reported.

3. Crystallographic and structural refinement parameters for reported crystals are required to be added in main text of manuscript.

Author Response

1)I carefully re-read the manuscript in its revised version and have -hopefully- eliminated all typos.

2) I agree that FT-IR, TGA and PXRD analysis would complete the characterization. However, we had noted that IR is generally very insensitive to variation in cations with these compounds. I have added the IR data for the purified compound 4. In all the other cases the single-crystalline material was not sufficient to provide more than the data already given. The crude products were always mixtures, but the IR data didn't look much different from the DATA of compound 4. PXRD data would help to show impurities, but the NMR and mass spectra provided these informations as well. so we did not regard it necessary to measure PXRD as well. TGA would certainly be very helpful regarding the pores in compound 4, but unfortunately, at the time of synthesis our instrument was broken for a long time. And I have to state, that I am retired for more than three years now and have neither an office nor lab space nor co-workers, and due to local safety regulations all the prepared substances had to be disposed, and thus it is impossible to provide any further analytical data...

3)The crystallographic data have been moved from the Supporting Information to the main text of the manuscript.

Reviewer 2 Report

Presented work by K. Sünkel and others describe the synthesis of cyano-containing bis-cyclopentadienes and their coordination compounds based on potassium and silver. The work can be recommended after some revisions.

In Schemes 1,2 for compounds 2-5 the cyclopentadienyl ring is better represented with delocalization of the negative charge (put the minus in the center of the ring and replace the double bonds with a dotted circle). 

Regarding coordination chemistry of potassium and silver. References should be made to other papers that discussed the coordination environment and its geometry, bond lengths, etc. 

Please use the descriptions in the papers as examples (DOI: 10.1002/ejic.201901202; DOI: 10.1002/ejic.202100553). Note that the distances between the nearest metal ions in the crystal package can be provided. In the description of the crystal packing structure it is necessary to specify the presence of coordination 1D chains or other types, as it is not specified in the text of the article. It is possible to single out one or two coordination chains and show them in different (most successful projections). 

Table 2 shows the angle -126.1(5) for the C3-C4-C4'-C3' bonds. This value is confusing because it is not clear what it refers to. 

Please use the value of the dihedral angle between the two planes in which the cyclopentadienyl rings are located.

Author Response

1)Schemes 1 and 2 have been re-drawn according to the reviewer's suggestions

2) Four new references have been added and discussed. I chose different ones than the two suggested by the reviewer, as those- although very interesting also- describe potassium coordination compounds of completely different ligand types, while the ones chosen by me contain Ag and K complexes of closely related polycyanoanion ligands. In part I do not understand what the reviewer means by "it is necessary ...as it is not specified in the text"- already in the original manuscript it was stated both in the "Results" and the "Discussion" section, that compounds 4 and 6 form 3D polymers. Anyway, I added some new Figures for the Supporting Information and a longer discussion of the coordination polymer.

3) In all tables I changed atom labels "X' " by the correct symmetry operator labels as shown in the Figures and their captions. It should be now clear which torsion angle was meant. In addition I added a sentence to the textual description, stating the equivalence of this torsion angle and the twist angle.

Reviewer 3 Report

The paper by Sünkel et al. presents high-quality research in the field of polynitrile coordination chemistry. The paper is very well written and the data presentation is clear. The only correction I would like to suggest is that separated atom charges should be added to the structure of 1 (Scheme 1, plus for N and minus for C) for correctness. 

Author Response

The structure drawing in Scheme 1 has been corrected according to the reviewer's suggestion.